# Pain after laparoscopic endometriosis-specific vs. hysterectomy surgeries: A retrospective cohort analysis

Yael Yagur[1⊙], Offra Engel[1⊙]*, Rachel Burstein[1], Justin Bsharat[2], Omer Weitzner[1], Yair Daykan[1], Zvi Klein[1], Ron Schonman[1]

**1** Meir Medical Center, Department of Obstetrics and Gynecology, Affiliated with the Faculty of Medicine, Tel Aviv University, Tel Aviv, Israel, **2** School of Medicine, New York State/American Program of Tel Aviv University, Tel Aviv, Israel

⊙ These authors contributed equally to this work.

* offra.engel@gmail.com

## Abstract

### Objectives

To evaluate pain perception and analgesic use between patients who underwent endometriosis-specific laparoscopic surgery compared to laparoscopic hysterectomy.

### Material and methods

This retrospective cohort study included women diagnosed with endometriosis who underwent laparoscopic surgery from 1/2019 to 11/2022. The control group consisted of premenopausal women who underwent laparoscopic hysterectomy, which was considered a similarly extensive surgery. Demographics, preoperative and post-operative data were compared between groups. Post-operative pain scores on a visual analogue scale (VAS) between 0 (no pain) and 10 (worst pain) were compared between groups for each post-operative day (POD). Standard pain relief analgesia on POD 0–1 included fixed intravenous treatment with paracetamol and intramuscular diclofenac. The need for additional analgesics (morphine or dipyrone) beyond the standard pain relief protocol was compared between groups.

### Results

Among 200 patients who underwent laparoscopic surgery, 100 (50%) were in the endometriosis group and 100 (50%) in the hysterectomy group. The endometriosis group was characterized by younger age and lower parity (both, p<0.001). There was no significant difference between the groups in mean VAS scores for each post-operative day. However, among patients who needed additional analgesics beyond the standard protocol on POD 1, a higher percentage of women in the endometriosis group used opioids rather than milder analgesics, as compared to controls (1% vs. 0.2%, respectively, p = 0.03).

**Data Availability Statement:** The data contains potentially identifying or sensitive patient information. The institutional review board does

not approve the publication of data in public databases, as it may inadvertently reveal the identities of patients who did not consent to participate in this retrospective study. Ensuring confidentiality is paramount to prevent any potential ethical concerns. The contact details of the Ethics Committee are: Email: meirhelsinki@clalit.org.il Phone: +972-9-7471588 Address: The Helsinki Committee Clinic Department, floor -1 Tschernichovsky 59 Kfar Saba Israel.

**Funding:** The authors received no specific funding for this work.

**Competing interests:** The authors have declared that no competing interests exist.

## Conclusion

Increased post-operative morphine use was observed in patients with endometriosis following laparoscopic surgery, despite no significant difference in mean VAS scores during the post-operative days. These findings suggest that personalized pain relief protocols should be adjusted for women with endometriosis.

## Introduction

Minimally invasive surgery is widely used in gynecological surgery [1, 2], particularly for the treatment of endometriosis [3]. This approach offers several advantages over traditional exploratory laparotomy, including reduced wound related pain, less analgesia use, minimal trauma, shorter hospital stays, faster recovery times, and earlier return to daily activities and work [1, 4, 5].

In the last decade, the enhanced recovery after surgery (ERAS) pathway has been employed as a strategy to effectively control post-operativepain after gynecological surgery [6]. The ERAS strategy utilizes multimodal analgesic regimens that use non-opioid analgesics (nonsteroidal anti-inflammatory drugs, cyclooxygenase inhibitors, acetaminophen, dipyrone, etc.) and supplemental opioid analgesics [7]. It recognizes the importance of optimal post-operative pain control to achieve other ERAS targets, which have been proven to reduce post-operative complications and expedite recovery [6].

Endometriosis is an estrogen-dependent inflammatory disease that affects approximately 10% of women of reproductive age [8]. One of the main characteristics of endometriosis is the variety of pain symptoms experienced, including dysmenorrhea, cyclic and acyclic lower abdominal pain, cyclic dysuria, dyschezia, and dyspareunia [9, 10]. The pathophysiology of pain in endometriosis comprises sensory and somatoform pain mechanisms [10]. The pain severity does not always correlate with the extent of endometriosis as classified by the revised American Fertility Society score/American Society for Reproductive Medicine scoring system [11].

Previous studies have demonstrated that women with endometriosis experience hypersensitivity to pain [12, 13], where even non-painful stimuli can evoke exaggerated pain perception [13]. This abnormal pain perception is a result of the chronic inflammatory process in endometriosis, defined as nociplastic pain which results in damage to the surrounding tissue [14, 15]. Over time, this inflammatory process, leading to decreased pain inhibition and amplified sensory input, can result in central sensitization [15, 16].These changes in pain processing in the central nervous systems have also been seen on experimental imaging [14] as functional and structural rearrangements of the anterior brain in women with endometriosis and chronic pelvic pain [17].

Patients experiencing both central sensitization and endometriosis commonly exhibit chronic pain, allodynia, hypersensitivity, and hyperalgesia [18], as well as mood disturbances. They typically demonstrate unsatisfactory responses to hormonal therapy [15].

The hypothesis of an association of the endometriosis type with the occurrence of central sensitization has only been confirmed for deep infiltrating lesions in the posterolateral parametria [15]. This leads to the assumption that other variables have greater impact on central sensitization. In some cases, this central sensitization can become independent of peripheral inputs, leading to long-term pain [19, 20].

Therefore, the generation of pain in endometriosis is a complex interplay of peripheral and central sensitization mechanisms, resulting in the generation of acute, chronic, cyclic and acyclic pain symptoms [21, 22].

Given the complexity and variety of pain mechanisms found in endometriosis patients, individualized pain management tailored to each patient is necessary [23]. Treatment options include medical (hormonal, non-hormonal, NSAIDs, opioids, etc.) and surgical approaches.

The aim of this study was to evaluate pain perception and analgesic use after minimally invasive laparoscopic surgery for endometriosis compared to that of patients who underwent laparoscopic hysterectomy

## Materials and methods

### Patients

This retrospective cohort study included women who underwent elective minimally invasive endoscopic surgery at the gynecology department of Meir Medical Center from 2019–2022. The study group included patients who underwent laparoscopic endometriosis surgery. Preoperative diagnosis was based on thorough clinical evaluation including physical examination and ultrasound data, with suspected diagnosis of stage 3–4 endometriosis according to the American Society for Reproductive Medicine Endometriosis Classification System. Women in this group underwent ovarian endometrioma cystectomy, adhesiolysis, and removal of endometriosis lesions from the ureter, sacro-uterine ligament, and the rectovaginal septum, and bowel shaving. The control group consisted of women who underwent simple laparoscopic hysterectomy with salpingectomy due to fibroid uterus, endometrial hyperplasia, or cervical intraepithelial neoplasia 3 (CIN3) during the same period as the study group. All surgeries were performed by the same team.

Women diagnosed with adenomyosis were excluded to prevent the inclusion of diagnoses in the control group to those of the study group. Patients with malignancies that indicated laparoscopic hysterectomy were also excluded because the surgeries were performed by a different surgical team specializing in gynecologic oncology. This exclusion was essential to maintain consistency in surgical teams across all procedures. Patients undergoing other procedures, such as myomectomy, cystectomy, salpingectomy, oophorectomy, or other benign gynecological procedures, were also excluded to ensure a clean, comparable control group and to minimize variations in surgical procedures that could potentially influence pain. Also excluded from the endometriosis group were patients whose pathological reports did not support the diagnosis and patients with intra and immediate postoperative complications due to bias the etiology of the of severe pain. Patients with diagnoses of other background diseases associated with chronic pain were also excluded.

### Surgical techniques

The study group included patients who underwent laparoscopic surgery for endometriosis. All procedures were performed by the same surgical team, with broad experience in endometriosis surgery. The same standard steps were followed for all surgeries. The abdomen was accessed by abdominal insufflation with a Veress needle or with a 10 mm umbilical trocar and three 5-mm accessory ports in the lower abdomen, under direct visualization. The next step is complete adhesiolysis followed by ovarian surgery, when needed. When deep parametrial endometriosis is diagnosed, the retroperitoneal area is accessed and the ureter identified. The medial and lateral pararectal spaces (Okabayashi and Latzko spaces) are developed based on the extent of the lesion. According to the involvement of the lesion, the ureter is lateralized and ureterolysis is performed, if required. The hypogastric nerves are preserved, if possible. In patients with

rectovaginal endometriosis nodules, the rectovaginal space is developed. First the nodule is shaved from the bowel and the last step is excision of the nodule from the vaginal wall with vaginal closure, if needed. None of the study patients required colectomy.

In the control group, all procedures included laparoscopic hysterectomy following the same standard steps. The abdomen was accessed by abdominal insufflation with a Veress needle or with a 10 mm umbilical trocar and three 5-mm accessory ports in the lower abdomen, under direct visualization or with an optical trocar. The surgical technique was the same for all patients using the following steps: coagulation and transection of both round ligaments, opening of the anterior fold of the broad ligament, developing the vesico-uterine space until exposing 2 cm of vaginal wall, coagulation and transection of the fallopian tube and ovarian ligaments, opening the posterior fold of the broad ligament up to the sacro-uterine ligaments, coagulation and transection of both sides of the uterine vessels and sacro-uterine ligaments, circular colpotomy and removal of the uterus through the vagina. The last step is laparoscopic closure the vaginal opening with a continuous barbed suture.

## Data

Data collected from electronic medical records included demographic information, menstrual characteristics (amenorrhea, irregular menstrual cycle, dysmenorrhea, menorrhagia), preoperative symptoms (urinary symptoms, dyspareunia, irregular bowel movements), preoperative examination findings (fibroid uterus, tenderness, frozen pelvis, presence of ovarian cyst/mass), and post-operative data (analgetic regimen including type, dosage and frequency), hemoglobin and white blood cell levels, duration of hospitalization, visual analogue scale (VAS) scores on each POD, and postoperative complications. The two groups were compared.

The post-operative pain perception was assessed using the VAS pain scores, ranging from 0 (0 no pain) and 10 (the worst pain), at each POD. These scores were compared between the two groups and the highest score reported for each POD was collected. In the initial data collection, every patient was coded to prevent identification. The team analyzing the results was exposed only to the coded data.

## Post-operative pain management

The standard pain protocol in our medical center, as part of the ERAS program for gynecological surgeries includes routine pain relief analgesia on post-operative day (POD) 0–1. This involves intravenous paracetamol 1,000 mg x 4/day and intramuscular diclofenac 75 mg x 2/day, with the addition of 40 mg of pantoprazole per day to prevent gastrointestinal irritation, unless the patient declines. If the patient requires additional analgesic treatment, we offer dipyrone 1000 mg up to 4 doses per day or subcutaneous morphine 5 mg up to a 3 times per day, based on the patient's reported VAS score and personal preferences.

We analyzed the patients' need for pain medication, including whether they followed the prescribed protocol, whether an additional medication was required, and if so, the type and dosage. Data regarding analgesic use was collected for POD 1 and POD 2, as analgesic use on POD 0 was not analyzed because patients are directed from the operating room to a recovery care unit. The standard pain protocol as part of the ERAS program begins when the patient arrives at the gynecology department after being released from the recovery unit during POD 0.

Patient characteristics and preoperative and post-operative data were compared between those who underwent laparoscopic endometriosis surgery and those who underwent laparoscopic hysterectomy.

A logistic regression model was constructed using the surgical procedure and preoperative diagnosis as dependent variables, and use of subcutaneous morphine as the independent

variable. The primary study outcome was the comparison of pain perception between the two groups based on VAS scores and post-operative analgesic use.

## Ethics

The study was approved by the local Institutional Review Board (#0197–22). The Ethics Committee waived the need for informed consent as the data were obtained retrospectively and fully anonymized.

## Statistical analysis

The study groups were compared post-operativeusing chi-square tests for categorical variables and independent t-tests for continuous variables. The mean VAS scores on each post-operative day and the mean dosages of analgesics used were compared between groups using t-tests. Logistic regression analyses were used to estimate odds ratios and 95% confidence intervals. A p-value less than 0.05 was considered statistically significant. All analyses were performed using SPSS, version 23 (IBM Corp., Armonk, NY, USA).

## Results

The study included 200 women who were admitted for minimally invasive gynecologic surgery and met the inclusion criteria. Among them, 100 (50%) were in the endometriosis group and 100 (50%) were in the control group. In the study group, 44 (88%) underwent excision of a sacro-uterine ligament endometriosis lesion with parametrial dissection, 28 (55%) excision of rectovaginal nodules, 36 (65%) treatment for ovarian endometrioma, and 5 (10%) excision of bladder endometriosis.

Baseline characteristics are presented in Table 1. The endometriosis group was characterized significantly by younger age and lower parity, while there were no significant differences in smoking and patient comorbidity between groups.

On physical examination, the study group had a higher prevalence of dysmenorrhea, dyspareunia, and alterations in bowel movements ($p<0.01$), as well as a higher prevalence of uterine tenderness (25% vs. 15% in the control group, p = 0.12), frozen pelvis (3% vs. 0, p = 0.1), and ovarian mass (58% vs. 25%, $p<0.01$) In contrast, the control group had a higher prevalence of menorrhagia (77% vs. 22% $p<0.01$), irregular menstrual cycles (41% vs. 9%, $p<0.01$), larger uterus on preoperative ultrasound (97.5 cm vs. 55.1 cm, $p<0.01$), and lower preoperative hemoglobin (11.4 mg/dl vs. 12.4 mg/dl, $p<0.01$).

There was no significant difference in mean VAS scores between the groups during the post-operative days (Table 2). However, a trend toward higher mean VAS scores was observed in the endometriosis group from POD 0 to POD 2. A between-subjects effect test showed that the differences in the dynamics of mean VAS scores reported on POD 2 were significant (using, p = 0.005).

Throughout the hospitalization period, more patients in the endometriosis group opted to use the full analgesic dosage allowed by the standard protocol plus additional doses, but the difference was not significant (Table 3). Among patients who required additional analgesics beyond the standard protocol on POD 1, a higher percentage of women in the endometriosis group used opioids instead of milder analgesics (dipyrone) compared to the control group (10% vs. 1%, respectively, p = 0.03). In comparison, the study group had a higher mean daily dosage of dipyrone, but the difference was not significant on POD 1 (1343 mg vs. 1292 mg in the control group, p = 0.6) and POD 2 (1230 mg vs. 1160 mg in the control group, p = 0.7). The medical analgesia characteristics of both groups are depicted in Table 3.

**Table 1. Sociodemographic and preoperative characteristics.**

| Characteristic | Endometriosis group | Control group | p-value |
|---|---|---|---|
| Age | 34.8±7.3 | 45.07 ±3.4 | <0.001 |
| Gravidity | 1.7±1.0 | 3.8 ±3.5 | <0.001 |
| Parity | 1.1 ±1.4 | 2.9± 1.6 | 0 |
| Smoking | 22 (22%) | 15 (15%) | 0.2 |
| Chronic illnesses | 29 (29%) | 47 (47%) | 0.029 |
| Psychological diagnosis | 4(4%) | 4 (4%) | 0.48 |
| **Menstrual cycle characteristics** | | | |
| Amenorrhea | 20 (20%) | 12 (12%) | PV לא סגורה על |
| Irregular cycle | | | PV לא סגורה על |
| Dysmenorrhea | 94 (94%) | 23 (23%) | 0.0000001 |
| Menorrhagia | 22 (22%) | 77 (77%) | 0.000001 |
| **Preoperative symptoms** | | | |
| Urinary symptoms | 19(19%) | 15 (15%) | 0.45 |
| Dyspareunia | 69 (69%) | 11 (11%) | 0.000001 |
| Alterations in bowel movements | 51 (51%) | 4(4%) | 0.000001 |
| **Preoperative findings** | | | |
| Fibroid uterus | 0 | 2 (2%) | |
| Tenderness | 25(25%) | 15 (15%) | 0.1 |
| Frozen pelvis | 3(3%) | 0 | 0.12 |
| Ovarian mass/cyst | 58 (58%) | 25 (25%) | 0.000001 |

There were no significant differences in the prevalence of post-operative complications such as post-operative infections (3% vs. 13% in the control group, p = 0.3). Multivariate logistic regression analysis revealed that morphine use on POD 1 was independently associated with more frequent use in the endometriosis group. (OR 4.8, 95% CI: 0–23, p = 0.04).

## Discussion

This study evaluated pain perception and analgesic use between patients who underwent endometriosis-specific laparoscopic surgery compared to laparoscopic hysterectomy. The results indicate that patients with endometriosis reported higher VAS scores during the post-operative period and had higher morphine use on POD 1 compared to the control group.

It is important to note that pain is a subjective experience that can be influenced by a variety of factors, including previous pain experiences, anxiety, and depression [24, 25]. Endometriosis is a known cause of chronic pelvic pain and can affect the pain perception of patients [26–28], leading to central hyperalgesia dysfunction [29, 30]. This dysfunction can cause either secondary allodynia or generalized hyperalgesia. These nociceptive classified nerves are found in the functional layer of the endometrium of women with endometriosis [12, 31]. These nerves

**Table 2. Mean VAS scores on each post-operative date.**

| VAS scores | Endometriosis group | Control group | p-value |
|---|---|---|---|
| POD 0 | 2.22 | 1.92 | 0.14 |
| POD 1 | 3.17 | 2.74 | 0.1 |
| POD 2 | 1.83 | 1.73 | 0.7 |

POD, post-operative day; VAS, visual analogue scale

**Table 3. Medical analgesia use characteristics.**

| | POD 1 | | | | | | POD 2 | | | | | |
|---|---|---|---|---|---|---|---|---|---|---|---|---|
| | Complied with pain protocol | Add on required | Type of add-on analgesic medicine | | | | Complied with pain protocol | Add on required | Type of add-on analgesic medicine | | | |
| | | | Dipyrone | | Morphine | | | | Dipyrone | | Morphine | |
| | | | Acquired | Mean dosage (mg) | Acquired | Mean dosage (mg) | | | Acquired | Mean dosage (mg) | Acquired | Mean dosage (mg) |
| **Control** | 59 (60.2%) | 36 (36.7%) | 36 (36%) | 1343.42 | 1 (1%) | 3.00 | 85 (86.7%) | 13 (13.3%) | 13 (13%) | 1230.77 | 0 | 0 |
| **Endometriosis** | 56 (57.1%) | 42 (42.9%) | 41 (41%) | 1292.68 | 10 (10%) | 7.30 | 13 (13.3%) | 19 (19.4%) | 19 (19%) | 1160.71 | 1 (1%) | 3 |
| **p-value** | 0.17 | 0.17 | 0.467 | 0.686 | 0.010 | 0.55 | 0.246 | לא סגורה על PV | 0.24 | 0.744 | 1 | |

are believed to have a significant role influencing central nervous system neurons and contributing to the perception of pelvic pain. The continuous activity of the nociceptors in the ectopic endometrium may cause hyper responsiveness of the neurons in the dorsal horn of the spinal cord and eventually result in central sensitization [31]. The sensitization is expressed as pain perception that is inappropriate to the time or degree of the primary lesion or injury [19] can be induced by loss of inhibitory synaptic transmission (usually mediated by GABA and glycerin receptors) or by increased excitatory transmission mediated by NMDA or AMPA receptors [32].

This study focused on the difference in surgical approaches between endometriosis-specific laparoscopies and laparoscopic hysterectomy and their effect on immediate postoperative pain. The main surgical difference between the two groups was the approach to the parametria. The surgical technique in patients with deep endometriosis lesions involving the parametrial area should consider the possibility of nerve involvement and nerve-sparing surgery. These spaces were not opened in other groups of patients; therefore, nerve involvement is less likely.

Rosati et al. described their technique for nerve-sparing radical hysterectomy in patients with deep parametrial endometriosis. Patients experienced significant improvements in dyschezia, dyspareunia and chronic pelvic pain [33, 34]. Ianieri et al. compared the influence of surgical treatment of endometriosis patients with and without involvement of the parametria on late postoperative symptoms. Other than higher risks of dyspareunia and sexual dysfunction in patients with parametrial involvement, the results were comparable for all symptoms [33, 34]. These data highlight the important investigating the immediate and short-term effects of parametrectomy on post-operative pain, further. We can only presume that these effects improve over time, as already demonstrated [34]. Immediate post-operative pain involves various aspects of the pain mechanism beyond the effect of nerve-sparing, which make these data less clear as a primary effect, although this may become apparent over time. We believe that in surgeries for deep infiltrating endometriosis involving parametrectomy, there is a greater extent of tissue damage, due to opening up more anatomical spaces and consequently triggering additional inflammatory healing processes in the immediate post-operative period. This heightened inflammation may contribute to increased post-operative pain and can also explain the opioid consumption; highlighting the need for an appropriate pain control protocol during this post-operative period. To substantiate this assumption, a more thorough evaluation of the influence of parametrial excision on immediate post-operative pain and pain management is warranted.

While NSAIDs may be sufficient for patients without chronic pelvic pain, the literature is inconsistent regarding their effectiveness for endometriosis-related pain [35].

This study provides further evidence that patients with endometriosis have an increased need for opioids following laparoscopic surgery compared to patients undergoing similar surgery.

The results of this study provide further evidence that patients following laparoscopic surgery for endometriosis have a higher need for opioids., which is consistent with previous reports. Delgado et al. [36] reported higher opioid use and more perceived post-operative pain in patients undergoing robotic surgery for endometriosis resection, and longer duration of opioid use.

One explanation for the higher requirement for opioid analgesia in endometriosis patients may be the role of neurogenic inflammation in the pathophysiology of the condition [37]. It has been shown that endogenous opioid peptides can produce analgesia by inhibiting the excitability of these nerves [37]. In contrast, Wong et al. [38] suggested that over-prescription by medical staff may be a contributing factor, particularly in patients with endometriosis, chronic pelvic pain, depression, or anxiety. They estimated that the average dose of opioids prescribed to endometriosis patients post-surgery was four times higher than needed.

Lazzeri et al. proposed an alternative mechanism, suggesting a correlation between disease severity and a heightened perception of stress, to the postoperative improvement [39]. Previous studies have explored the link between severe pain and deep endometriosis [40], along with a correlation indicating lower pain levels in cases of endometriomas compared to other locations [41]. However, perceived pain (dysmenorrhea, pelvic pain, and dyspareunia as assessed by the Perceived Stress Scale), was significantly ameliorated one month post-surgery in individuals with deep/infiltrating endometriosis, in contrast to those with peritoneal endometriosis or endometriomas only [39, 42]. The most symptomatic women reported pain relief, improved feelings of depression and stress perception after surgery. This could be another explanation for the similar VAS scores during the early post-operative days evaluated in both groups, because 73% of the study group patients had deep/infiltrating endometriosis. Further studies investigating immediate postoperative pain perception may lend support to this hypothesis, illustrating the persistent course of pain relief from the immediate post-operative phase to the later stages, as demonstrated by Lazzeri et al. one month after surgery (Fauconnier and Chapron 2005; Lazzeri et al. 2015)

It has been previously noted that uncontrolled post-operative pain can lead to post-operative complications, chronic post-operative pain [43], and dissatisfaction [44]. Although there are multiple options for post-operative pain management, each with its own benefits and drawbacks, current guidelines do not address specific issues such as changes in pain perception [26, 27, 35], preoperative chronic analgesic use [45, 46], or the unique mechanisms of pain in patients with endometriosis [26]. Based on the findings of this study and previous literature, we recommend a more aggressive, standard, pain management protocol for patients scheduled for laparoscopic resection of endometriosis, to prevent uncontrolled postoperative pain and associated complications, while reducing the need for opioid-based analgesia as advised by the ERAS protocol [6].

As mentioned, there were significant differences in the baseline characteristics of the study and control groups. These differences were a result of the natural course of the pathologies leading to surgical treatment. Although there were differences in age, parity, pre-operative symptoms and physical findings, the main variables that influenced analgesic consumption were similar between the two groups. Both groups underwent laparoscopic access with the same intraabdominal pressure range, similar extent of the operation, duration and complexity, and received similar anesthesia. Potential complications were similar, as well [47, 48]. It seems that factors that influence pain perception in patients with endometriosis are the reason for the difference in analgesic treatment and not the extent of surgery.

The strengths of this study include using a consistent surgical technique performed by a small, specialized team. The use of the pain protocol was documented systematically and accurately by the nursing team. However, there are limitations to the study, such as the relatively small sample size, limited follow-up to the immediate post-operative period, and the natural differences in surgeries between the two groups. Additionally, all patients received a standard pain protocol, potentially masking differences in pain perception.

In conclusion, our findings suggest that patients undergoing laparoscopic resection of endometriosis lesions experience higher levels of post-operative pain compared to patients undergoing laparoscopic hysterectomy, and require more opioid-based pain medication. Given the negative impact of uncontrolled post-operative pain, a customized analgesic regimen may be necessary for these patients. Further research is needed to confirm and expand on these findings.

## Author Contributions

**Conceptualization:** Yael Yagur, Offra Engel, Omer Weitzner, Zvi Klein, Ron Schonman.

**Data curation:** Yael Yagur, Rachel Burstein.

**Formal analysis:** Yael Yagur, Offra Engel, Omer Weitzner.

**Investigation:** Yael Yagur, Offra Engel, Omer Weitzner, Zvi Klein, Ron Schonman.

**Methodology:** Yael Yagur, Zvi Klein, Ron Schonman.

**Project administration:** Offra Engel, Zvi Klein.

**Supervision:** Yael Yagur, Yair Daykan, Ron Schonman.

**Validation:** Yael Yagur, Rachel Burstein, Justin Bsharat, Yair Daykan, Zvi Klein, Ron Schonman.

**Writing – original draft:** Offra Engel.

**Writing – review & editing:** Yael Yagur, Offra Engel, Justin Bsharat, Omer Weitzner, Yair Daykan, Zvi Klein, Ron Schonman.

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
