## [Decision Letter · Decision Letter 0]

26 Dec 2023

PONE-D-23-18897Differences in post operative pain perception among patients with endometriosisPLOS ONE

Dear Dr. Engel,

Thank you for submitting your manuscript to PLOS ONE. After careful consideration, we feel that it has merit but does not fully meet PLOS ONE’s publication criteria as it currently stands. Therefore, we invite you to submit a revised version of the manuscript that addresses the points raised during the review process.

We look forward to receiving your revised manuscript.

Kind regards,

Diego Raimondo

Academic Editor

PLOS ONE

Journal Requirements:

2. "We noticed you have some minor occurrence of overlapping text with the following previous publication(s), which needs to be addressed:

https://www.annualreviews.org/doi/10.1146/annurev-physiol-012110-142158

https://doaj.org/article/3ec5ad6e982449348e3502c132fb1044

In your revision ensure you cite all your sources (including your own works), and quote or rephrase any duplicated text outside the methods section. Further consideration is dependent on these concerns being addressed

Reviewers' comments:

Reviewer's Responses to Questions

**Comments to the Author**

1. Is the manuscript technically sound, and do the data support the conclusions?

Reviewer #1: Partly

Reviewer #2: Yes

2. Has the statistical analysis been performed appropriately and rigorously? 

Reviewer #1: Yes

Reviewer #2: Yes

3. Have the authors made all data underlying the findings in their manuscript fully available?

Reviewer #1: Yes

Reviewer #2: Yes

4. Is the manuscript presented in an intelligible fashion and written in standard English?

Reviewer #1: No

Reviewer #2: Yes

5. Review Comments to the Author

Reviewer #1: Dear Authors,

I suggest some amendments in order to obtain a more enjoyable paper. Currently, the manuscript has some serious flaws that should be addressed.

1- first, I suggest you to focus on what the manuscript investigates, which is not the postoperative use of analgesics according to the presence of endometriosis or not, but the different surgical approaches (endometriosis specific surgical procedures vs hysterectomy) and their effect on pain. This confounding is the main problem of the paper and you fall with it throughout all your article, starting from the introduction up to the discussion. Otherwise, please clearly explain.

2- accordingly, you should rephrase the title.

3- please cite and report in the introduction this recent and interesting article about central sensitization and risk factors: https://doi.org/10.1016/j.jmig.2022.10.007

4- Why does the control group do not include endometrial cancer or other diseases?

5- Please define "had similar surgical and hospitalization characteristics, such as surgery duration and complexity, postoperative recovery, and potential complications". What do you mean with similar? Is this an objective definition?

6- Did all the surgeons have the same surgical experience?

7- Material and methods section need a grammar revision.

8- enhance discussion on how surgery affects pain perception in endometriosis patients (https://doi.org/10.1016/j.jmig.2015.08.639)

9- rearrange discussion accordingly to point number 1.

Reviewer #2: The manuscript proposed by the Author appears to be original and interesting, but to be accepted for publication, some sections of the study must be improved and implemented:

-In the introduction I suggest focusing better on the concept of pain sensitization by also citing works such as this "Prevalence and Risk Factors of Central Sensitization in Women with Endometriosis", Raimondo D et al. 2023

-in the materials and methods section there is no description of the surgical technique and above all the extent of surgery for endometriosis (were visceral resections performed? Parametrectomies?). These aspects must therefore also be specified in a descriptive table in the "results" section

-The discussion section should be significantly expanded, also discussing the role that the surgical procedures performed in the group of patients operated for endometriosis may have had in the use of any opioids in the post-operative period. It is important in this section to discuss the possible role of parametrectomy in endometriosis surgery ("Impact of nerve-sparing posterolateral parametrial excision for deep infiltrating endometriosis on postoperative bowel, urinary, and sexual function", Ianieri et al 2022) and of modified radical hysterectomy with parametrectomies for endometriosis (“Surgical and functional impact of nerve-sparing radical hysterectomy for parametrial deep endometriosis: a single center experience”, Rosati A et al, 2022).

Furthermore, it could be interesting to understand whether patients with functional disorders linked to pelvic floor disfunction may need greater use of pain-relieving therapy in the post-operative period ("Pelvic floor dysfunction at transperineal ultrasound and chronic constipation in women with endometriosis", Raimondo D et al 2022)

6. PLOS authors have the option to publish the peer review history of their article (what does this mean?). If published, this will include your full peer review and any attached files.

Reviewer #1: No

Reviewer #2: No

---

## [Author Response · Author response to Decision Letter 0]

27 Feb 2024

Reviewer #1:

1-First, I suggest you to focus on what the manuscript investigates, which is not the postoperative use of analgesics according to the presence of endometriosis or not, but the different surgical approaches (endometriosis specific surgical procedures vs hysterectomy) and their effect on pain. This confounding is the main problem of the paper and you fall with it throughout all your article, starting from the introduction up to the discussion. Otherwise, please clearly explain.

RESPONSE: Thank you for your comment. We agree with your clarification and changed the concept as you mentioned.

Abstract: Line 23 

Objectives: : To evaluate pain perception and analgesic use between patients who underwent endometriosis-specific laparoscopic surgery compared to laparoscopic hysterectomy 

Introduction: Line 103 

The aim of this study was to evaluate pain perception and analgesic use after minimally invasive laparoscopic surgery for endometriosis compared to patients who underwent laparoscopic hysterectomy.

Discussion: Line 340

 As we understand the complexity of pain perception mechanisms involved in endometriosis, we wished to explore and compare postoperative pain management for patients who underwent endometriosis surgery to that of patients following laparoscopic hysterectomy only. This enables a better understanding of the need for a tailored approach to postoperative pain management following laparoscopic endometriosis surgery. In this study, we demonstrated that anti-inflammatory analgesics had a suboptimal effect on acute post-operative pain following laparoscopic surgery for removal of endometriosis lesions. 

Line 354: This study provides further evidence that patients have higher need for opioids following laparoscopic surgery for endometriosis compared to patients after laparoscopic hysterectomy. 

Line 394: Based on the findings of this study and previous literature, we recommend a more aggressive fixed protocol for pain management for patients undergoing laparoscopic resection of endometriosis lesions, to prevent uncontrolled postoperative pain and associated complications, while reducing the need for opioid-based analgesia as advised by the ERAS protocol (6).

Line 419: In conclusion, our findings suggest that patients undergoing laparoscopic resection of endometriosis lesions experience higher levels of postoperative pain compared to patients undergoing laparoscopic hysterectomy, and require additional opioid-based pain medication. Given the negative impact of uncontrolled postoperative pain, a customized analgesic regimen may be necessary for these patients. Further research is needed to confirm and expand on our results.

2- Accordingly, you should rephrase the title.

RESPONSE: Revised to "Comparative Analysis of Pain Outcomes in Laparoscopic Endometriosis-Specific Surgery vs. Hysterectomy" 

Pain after laparoscopic endometriosis-specific vs. hysterectomy surgeries: A retrospective cohort analysis

3- Please cite and report in the introduction this recent and interesting article about central sensitization and risk factors: https://doi.org/10.1016/j.jmig.2022.10.007

RESPONSE: Thank you for this comment. The research conducted by Raimondo et al. are important, and its implications have been incorporated (reference 15) as requested.

Line 76-95: 

Previous studies have demonstrated that women with endometriosis experience hypersensitivity to pain (12,13), where even non-painful stimuli can evoke exaggerated pain perception (13). This abnormal pain perception is a result of the chronic inflammatory process in endometriosis, defined as nociplastic pain which results in damage to the surrounding tissue (14)(15). Over time, this inflammatory process, leading to decreased pain inhibition and amplified sensory input, can result in central sensitization (15,16). These changes in pain processing in the central nervous systems have also been seen on experimental imaging (14) as functional and structural rearrangements of the anterior brain in women with endometriosis and chronic pelvic pain (17). 

Patients experiencing both central sensitization and endometriosis commonly exhibit chronic pain, allodynia, hypersensitivity, and hyperalgesia, (18) as well as mood disturbances. They typically demonstrate unsatisfactory responses to hormonal therapy(15).

The hypothesis of an association of the endometriosis type with the occurrence of central sensitization has only been confirmed for deep infiltrating lesions in the posterolateral parametria (15). This leads to the assumption that other variables have greater impact on central sensitization.

4- Why does the control group do not include endometrial cancer or other diseases?

RESPONSE: Thank you for providing the opportunity to clarify. The decision to exclude endometrial cancer or other diseases from the control group was intentional and aimed at maintaining consistency in surgical teams across all procedures. Our team focuses exclusively on benign gynecological diseases, the surgeons performing oncological surgeries constitute a distinct team. This approach allowed us to use the same team for both hysterectomy and endometriosis surgeries. 

We added to the Materials and Methods section, Lines 132-143:

Women diagnosed with adenomyosis were excluded to prevent the inclusion of diagnoses in the control group to those of the study group. Patients with malignancies that indicated laparoscopic hysterectomy were also excluded because the surgeries were performed by a different surgical team specializing in gynecologic oncology. This exclusion was essential to maintain consistency in surgical teams across all procedures. Patients undergoing other procedures, such as myomectomy, cystectomy, salpingectomy, oophorectomy, or other benign gynecological procedures, were also excluded to ensure a clean, comparable control group and to minimize variations in surgical procedures that could potentially influence pain. Also excluded from the endometriosis group were patients whose pathological reports did not support the diagnosis and patients with intra and immediate postoperative complications due to bias the etiology of the of severe pain . Patients with diagnoses of other background diseases associated with chronic pain were also excluded.

5- Please define "had similar surgical and hospitalization characteristics, such as surgery duration and complexity, postoperative recovery, and potential complications". What do you mean with similar? Is this an objective definition?

RESPONSE: To increase the comparability of the groups, we aimed to create the most uniform selection possible, excluding factors that could influence analgesic use. The primary approach involved ensuring the surgical procedures were as consistent as possible. In the study group, we included patients with endometriosis type 3-4 with surgical procedures including excision of deep endometriosis lesions, ovarian endometriomas, cystectomy and excision of rectovaginal lesions. The control group comprised patients who underwent laparoscopic hysterectomy for fibroid uterus, endometrial hyperplasia, or cervical intraepithelial neoplasia 3. Patients experiencing complications during or after surgery were excluded, as were those with other chronic pain diseases. 

We decided to delete this paragraph because we believe that other sections, which provide additional information about the surgical team, procedures, and group selection, offer clearer explanations.

We added a paragraph detailing the surgical steps undertaken in each group to the Material and Methods section. 

Lines 148-172: “The study group included patients who underwent laparoscopic surgery for endometriosis. All procedures were performed by the same surgical team, with broad experience in endometriosis surgery. The same standard steps were followed for all surgeries. The abdomen was accessed by abdominal insufflation with a Veress needle or with a 10 mm umbilical trocar and three 5-mm accessory ports in the lower abdomen, under direct visualization. The next step is complete adhesiolysis followed by ovarian surgery, when needed. When deep parametrial endometriosis is diagnosed, the retroperitoneal area is accessed and the ureter identified. The medial and lateral pararectal spaces (Okabayashi and Latzko spaces) are developed based on the extent of the lesion. According to the involvement of the lesion, the ureter is lateralized and ureterolysis is performed, if required. The hypogastric nerves are preserved, if possible. In patients with rectovaginal endometriosis nodules, the rectovaginal space is developed. First the nodule is shaved from the bowel and the last step is excision of the nodule from the vaginal wall with vaginal closure, if needed. None of the study patients required colectomy.

In the control group, all procedures included laparoscopic hysterectomy following the same standard steps. The abdomen was accessed by abdominal insufflation with a Veress needle or with a 10 mm umbilical trocar and three 5-mm accessory ports in the lower abdomen, under direct visualization or with an optical trocar. The surgical technique was the same for all patients using the following steps: coagulation and transection of both round ligaments, opening of the anterior fold of the broad ligament, developing the vesico-uterine space until exposing 2 cm of vaginal wall, coagulation and transection of the fallopian tube and ovarian ligaments, opening the posterior fold of the broad ligament up to the sacro-uterine ligaments, coagulation and transection of both sides of the uterine vessels and sacro-uterine ligaments, circular colpotomy and removal of the uterus through the vagina. The last step is laparoscopic closure the vaginal opening with a continuous barbed suture. 

6- Did all the surgeons have the same surgical experience?

RESPONSE: Our surgical teams share a common focus on benign gynecological diseases, each team is planned to specialize in different procedures. This intentional allocation allowed us to maintain consistency within each team for specific surgeries. 

While our surgical team comprises individuals with varying levels of experience, we ensured consistency by maintaining the same team for all surgeries under the guidance of a very experienced and well-trained lead surgeon, who specializes in advanced laparoscopic surgeries. This was planned to ensure consistency across all surgeries.

Our emphasis on maintaining the same team for specific surgeries was intended to minimize variability within comparable groups, enabling a more accurate analysis of the impact of different procedures on pain outcomes.

7- Material and methods section need a grammar revision.

RESPONSE: Thank you for your comment, which was well taken. The manuscript was reviewed by a native English speaker who is a medical editor.

8- Enhance discussion on how surgery affects pain perception in endometriosis patients (https://doi.org/10.1016/j.jmig.2015.08.639)

RESPONSE: Thank you for this comment, the Discussion was expanded according to your suggestion . 

Lines 362-377: Lazzeri et al. proposed an alternative mechanism, suggesting a correlation between disease severity and a heightened perception of stress, to the postoperative improvement (37). Previous papers have explored the link between severe pain and deep endometriosis (38), along with a correlation indicating lower pain levels in cases of endometrioma compared to other locations (39). However, the perceived pain (dysmenorrhea, pelvic pain and dyspareunia, as assessed by the Perceived Stress Scale), was significantly ameliorated one month post-surgery in individuals with deep/infiltrating endometriosis, in contrast to those with peritoneal endometriosis or endometriomas only (Lazzeri et al. 2015) (40). The most symptomatic women (preoperatively) reported pain relief, and improved feelings of depression and stress perception after surgery. This could be another explanation for the similar VAS scores during the early postoperative days in both groups, because 73% of the study group patients had deep/infiltrating endometriosis. Additional studies investigating immediate postoperative pain perception may lend support to this hypothesis, illustrating the persistent course of pain relief from the immediate postoperative phase to the later stages, as demonstrated by Lazzeri et al. one month after surgery

9- Rearrange discussion accordingly to point number 1.

RESPONSE: The discussion was rearranged, as suggested. 

Discussion, Line 332: 

As we understand the complexity of pain perception mechanisms involved in endometriosis, we explored and compared postoperative pain management for patients who underwent endometriosis surgery to that of patients following laparoscopic hysterectomy only. This enables a better understanding of the need for a tailored approach to post-operative pain management following laparoscopic endometriosis surgery. in the results of this study indicate that anti-inflammatory analgesics had suboptimal effect in treating acute postoperative pain following laparoscopic surgery for removal of endometriosis lesions. 

Line 342: The results of this study provide further evidence that patients following laparoscopic surgery for endometriosis have a higher need for opioids. 

Line 378: Based on the findings of this study and previous literature, we recommend a more aggressive, standard, pain management protocol for patients scheduled for laparoscopic resection of endometriosis, to prevent uncontrolled postoperative pain and associated complications, while reducing the need for opioid-based analgesia as advised by the ERAS protocol (6).

Line 403: In conclusion, our findings suggest that patients undergoing laparoscopic resection of endometriosis lesions experience higher levels of postoperative pain compared to patients undergoing laparoscopic hysterectomy, and require more opioid-based pain medication. Given the negative impact of uncontrolled postoperative pain, a customized analgesic regimen may be necessary for these patients. Further research is needed to confirm and expand on these findings.

Reviewer #2

The manuscript proposed by the Author appears to be original and interesting, but to be accepted for publication, some sections of the study must be improved and implemented:

-In the introduction I suggest focusing better on the concept of pain sensitization by also citing works such as this "Prevalence and Risk Factors of Central Sensitization in Women with Endometriosis", Raimondo D et al. 2023

RESPONSE: Thank you for this comment. The research conducted by Raimondo et al. is important, and its implications have been incorporated in the Introduction section, as requested. 

Line 76-95:

Previous studies have demonstrated that women with endometriosis experience hypersensitivity to pain (12,13), where even non-painful stimuli can evoke exaggerated pain perception (13). This abnormal pain perception is a result of the chronic inflammatory process in endometriosis, defined as nociplastic pain, which results in damage to the surrounding tissue (14)(15). Over time, this inflammatory process, leading to decreased pain inhibition and amplified sensory input, can result in central sensitization (15,16). These changes in the pain processing in central nervous system of women with endometriosis have been also shown on experimental imaging (14), with functional and structural rearrangements of the rostral structures in women with endometriosis and chronic pelvic pain (17). 

Patients experiencing both central sensitization and endometriosis commonly exhibit chronic pain, allodynia, hypersensitivity, hyperalgesia (18), as well as mood disturbances. They typically demonstrate unsatisfactory responses to hormonal therapy (15).

The hypothesis of an association between the endometriosis type and the occurrence of central sensitization was not confirmed in general (15), except for deep infiltrating lesions in posterolateral parametria (15). This leads to the presumption that there are other variables with greater impact on central sensitization.

-In the materials and methods section there is no description of the surgical technique and above all the extent of surgery for endometriosis (were visceral resections performed? Parametrectomies?). These aspects must therefore also be specified in a descriptive table in the "results" section

RESPONSE: We added the following to the Material and Methods section. 

Lines 148-172: The study group included patients who underwent laparoscopy for treatment of endometriosis. All laparoscopic endometriosis procedures were performed by the same surgical team with broad experience in endometriosis surgery, using the same standard steps. The abdomen was accessed by abdominal insufflation with a Veress needle or with a 10 mm umbilical trocar and 3 5-mm accessory ports in the lower abdomen under direct visualization. The surgery begins with complete adhesiolysis followed by ovarian surgery, when needed. When parametrial deep endometriosis involvement is diagnosed, the retroperitoneal area is accessed, the ureter is identified. The relevant spaces are developed according to the extension of the lesion; the medial and lateral pararectal spaces (Okabayashi and Latzko spaces). According to the involvement of the lesion, the ureter is lateralized and ureterolysis is performed, if needed. The hypogastric nerves are preserved if possible. In patients with rectovaginal endometriosis nodules, the rectovaginal space is developed. First the nodule is shaved from the bowel and the last step is excision of the nodule from the vaginal wall with vaginal closure, if needed. None of the study patients needed colectomy.

In the control group, all procedures included laparoscopic hysterectomy using the same standard steps. The abdomen was accessed by abdominal insufflation with a Veress needle or with a 10 mm umbilical trocar and 3 5-mm accessory ports in the lower abdomen under direct visualization. The surgical technique was the same for all patients using the following steps: coagulation and transection of both round ligaments, opening of the anterior fold of the broad ligament, developing the vesico-uterine space until exposing 2 cm of vaginal wall, coagulation and transection of the fallopian tube and ovarian ligaments, opening the posterior fold of the broad ligament up to the sacro-uterine ligament, coagulation and transection of both sides of the uterine vessels and sacro-uterine ligaments, circular colpotomy and removal of the uterus through the vagina. The last step is laparoscopic closure the vaginal opening with a continuous barbed suture.

We added a description of the surgical intervention for the endometriosis group to the Results section. Since the control group underwent laparoscopic hysterectomy only, we preferred to add this description to the text. 

Lines 237-240: “In the study group, 88% underwent excision of a sacro-uterine ligament endometriosis lesion with parametrial dissection, 55% excision of rectovaginal nodules, 65% treatment of ovarian endometrioma and 10% excision of bladder endometriosis.” 

-The discussion section should be significantly expanded, also discussing the role that the surgical procedures performed in the group of patients operated for endometriosis may have had in the use of any opioids in the post-operative period. It is important in this section to discuss the possible role of parametrectomy in endometriosis surgery ("Impact of nerve-sparing posterolateral parametrial excision for deep infiltrating endometriosis on postoperative bowel, urinary, and sexual function", Ianieri et al 2022) and of modified radical hysterectomy with parametrectomies for endometriosis (“Surgical and functional impact of nerve-sparing radical hysterectomy for parametrial deep endometriosis: a single center experience”, Rosati A et al, 2022).

RESPONSE: Thank you for the important comment. We added to the Discussion, 

Lines 303-329:

 “This study focused on the difference in surgical approaches between endometriosis-specific laparoscopies and laparoscopic hysterectomy and their effect on immediate postoperative pain. The main surgical difference between the two groups was the approach to the parametria. The surgical technique in patients with deep endometriosis lesions involving the parametrial area should consider the possibility of nerve involvement and the possibility of nerve sparing surgery while in other groups we did not open those spaces; therefore, nerve involvement is less likely. 

Rosati et al described their technique for nerve-sparing radical hysterectomy in patients with parametrial involvement of deep endometriosis. Patients experienced significant improvements in dyschezia, dyspareunia and chronic pelvic pain (Lanieri et al. 2022). Lanieri et al compared the effect of the surgical treatment of endometriosis patients with and without involvement of the parametria on late postoperative symptoms. Their results were comparable for all symptoms, except a higher risk of dyspareunia and sexual dysfunction in patients with parametrial involvement (Ianieri et al. 2022; Rosati et al. 2022). These data highlight the significance of further investigation into the immediate and short-term effects of parametrectomy on postoperative pain. We can only presume that these effects improve greatly over time, as already demonstrated. Immediate postoperative pain involves various pain mechanisms beyond the effect of nerve sparing surgery, which make these data less clear, as a primary effect and may become apparent over time. We believe that in surgeries involving parametrectomy for deep infiltrating endometriosis, tissue damage is greater, more anatomical spaces are opened, consequently triggering more inflammatory healing processes in the immediate postoperative period. This heightened inflammation may contribute to increased postoperative pain and can also explain the opioid consumption, highlighting the need for an appropriate pain control protocol during the postoperative period. To substantiate our assumption, a more thorough evaluation of the influence of parametrial excision on immediate postoperative pain and pain management is warranted.

- Furthermore, it could be interesting to understand whether patients with functional disorders linked to pelvic floor disfunction may need greater use of pain-relieving therapy in the post-operative period ("Pelvic floor dysfunction at transperineal ultrasound and chronic constipation in women with endometriosis", Raimondo D et al 2022)

RESPONSE: Thank you for highlighting the aspect of pelvic floor dysfunction and its potential impact on postoperative pain management. We acknowledge the significance of this question and its potential as a confounding factor. Although preoperative examinations specifically addressing pelvic floor dysfunction are not available for this study, we consider it a crucial aspect for future investigations. Recognizing its importance, we have incorporated this concern into our preoperative discussions, emphasizing the need for comprehensive exploration through imaging and physical examinations for subsequent patients.

---

## [Editor Report · Decision Letter 1]

11 Mar 2024

Pain after laparoscopic endometriosis-specific vs. hysterectomy surgeries: A retrospective cohort analysis

PONE-D-23-18897R1

Dear Dr. Engel,

We’re pleased to inform you that your manuscript has been judged scientifically suitable for publication and will be formally accepted for publication once it meets all outstanding technical requirements.

Kind regards,

Diego Raimondo

Academic Editor

PLOS ONE

Additional Editor Comments (optional):

the authors replied to all Reviewers’ queries. The manuscript provides new concepts and data for endometriosis research.

---

## [Editor Report · Acceptance letter]

20 Mar 2024

PONE-D-23-18897R1 

PLOS ONE

Dear Dr. Engel, 

I'm pleased to inform you that your manuscript has been deemed suitable for publication in PLOS ONE. Congratulations! Your manuscript is now being handed over to our production team.

Kind regards, 

on behalf of

Dr. Diego Raimondo 

Academic Editor

PLOS ONE